

# Melatonin improves the germination rate of cotton seeds under drought stress by opening pores in the seed coat

Yandan Bai[1], Shuang Xiao[1], Zichen Zhang[1], Yongjiang Zhang[1], Hongchun Sun[1], Ke Zhang[1], Xiaodan Wang[2], Zhiying Bai[3], Cundong Li[1] and Liantao Liu[1]

[1] College of Agronomy, HeBei Agricultural University/ State Key Laboratory of North China Crop Improvement and Regulation/ Key Laboratory of Crop Growth Regulation of Hebei Province, Baoding, Hebei Province, China, Baoding, China
[2] College of Traditional Chinese Veterinary Medicine, Hebei Agricultrual University, Baoding, China
[3] College of Life Science, Hebei Agricultrual University, Baoding, China

Corresponding author
Liantao Liu, liult@hebau.edu.cn

## ABSTRACT

The germination of cotton (*Gossypium hirsutum* L.) seeds is affected by drought stress; however, little is known about the physiological mechanism affecting germination and the effect of melatonin (MT) on cotton seed germination under drought stress. Therefore, we studied the effects of exogenous MT on the antioxidant capacity and epidermal microstructure of cotton under drought stress. The results demonstrated a retarded water absorption capacity of testa under drought stress, significantly inhibiting germination and growth in cotton seeds. Drought stress led to the accumulation of reactive oxygen species (ROS), malondialdehyde (MDA), and osmoregulatory substances (e.g., proline, soluble protein, and soluble sugars); it also decreased the activity of antioxidant enzymes and $\alpha$-amylase. Drought stress inhibited gibberellin acid ($GA_3$) synthesis and increased abscisic acid (ABA) content, seriously affecting seed germination. However, seeds pre-soaked with MT (100 $\mu$M) showed a positive regulation in the number and opening of stomata in cotton testa. The exogenous application of MT increased the germination rate, germination potential, radical length, and fresh weight, as well as the activities of superoxide dismutase (SOD), peroxidase (POD), and $\alpha$-amylase. In addition, MT application increased the contents of organic osmotic substances by decreasing the hydrogen peroxide ($H_2O_2$), superoxide anion ($O_2^-$), and MDA levels under drought stress. Further analysis demonstrated that seeds pre-soaked with MT alleviated drought stress by affecting the ABA and $GA_3$ contents. Our findings show that MT plays a positive role in protecting cotton seeds from drought stress.

## INTRODUCTION

Drought is a global challenge because of climate warming and increasingly constrained water resources (*Jury & Vaux, 2005*). Drought is also one of the main abiotic stress factors limiting the growth and yield of plants (*Farooq et al., 2009a*), and the changing global climate is making this situation more extreme (*Farooq et al., 2013*). The response of plants
to drought stress are interconnected (*Huang et al., 2019*); specifically, drought influences many morphological, physiological, and biochemical processes such as stomatal closure, cellular dehydration, the peroxidation of membrane lipids, and reduced antioxidant capacity, affecting plant survival and development (*Ahammad et al., 2019a*; *Huang et al., 2019*; *Gao et al., 2018*). However, plants have evolved complex stress-resistant mechanisms to counteract damage due to drought, which may allow plants to complete their life cycles (*Ahammed et al., 2019a*).

Germination is a complex process regulated by a series of physical and metabolic events in the life cycle of plants (*Léonie & Maarten, 2008*; *Xiao et al., 2019*). As the initial stage of plant life cycles and the critical stage of seedling establishment, seed germination is extremely sensitive to drought stress (*Ibrahim, 2016*). A previous study has shown that the mechanical barrier provided by the seed coat increases under stress conditions, indicating that drought stress inhibits seed germination (*Debeaujon, Léon-Kloosterziel & Koornneef, 2000*). Drought stress leads to the over-production of reactive oxygen species (ROS) in plant cells, which in turn triggers oxidative damage and may lead to cell death (*Farooq et al., 2009b*; *Nahar et al., 2015*; *Xiao et al., 2019*). However, plants have evolved efficient antioxidative defense mechanisms, including the activities of antioxidant enzymes and the regulation of non-enzymatic antioxidants to reduce ROS induced oxidative damages (*Posmyk et al., 2009*). Plant hormones are important signaling molecules that respond to environmental changes in seed germination (*Yamaguchi, Kamiya & Nambara, 2007*). Gibberellins (GA) and abscisic acid (ABA) are well-known phytohormones that play crucial roles in seed germination and early seedling establishment (*Shu et al., 2018*; *Ahammed et al., 2020a*). ABA, a universal abiotic stress hormone, responded positively to abiotic stress, while GA acts as a plant growth regulator to promote seed germination in response to stress (*Shu et al., 2018*). Drought stress adversely affects seed germination by disrupting the dynamic balance of ABA catabolism and GA biosynthesis (*Vishal & Kumar, 2018*; *Ahammed et al., 2020a*). A previous study reported the effects of GA and salinity on physiological characteristics in maize, and their findings showed that GA could reduce proline accumulation and electrolyte leakage, alleviating the damage to plants caused by salt stress by delaying the decrease of superoxide dismutase (SOD) and peroxidase (POD) (*Tuna et al., 2008*). Previous studies have shown that the accumulation of ABA is closely related to the generation of ROS; specifically, when the accumulation of ABA was lower, the formation of hydrogen peroxide ($H_2O_2$) reduced accordingly (*Liu et al., 2010*; *Ye et al., 2011*). Drought stress negatively regulates metabolism due to low α-amylase activity, leading to a significant inhibition of seed germination and seedling growth (*Farooq et al., 2020*). The accumulation of osmoregulatory substances, such as proline, soluble sugar, and soluble protein, which can reduce the water potential and enhance the water absorption capacity of cells, ensures normal plant metabolism and improves drought resistance (*Zhang et al., 2015a*; *Zhang et al., 2015b*).

Cotton, the world's major fiber and oil crop, is also severely affected by drought. Drought weakens seedling growth by limiting seed germination, which adversely affects cotton productivity (*Farah et al., 2012*; *Luo et al., 2018*). Therefore, it is urgent to improve the rate of cotton seed germination under drought conditions.

Melatonin (N-acetyl-5-methoxytryptamine, MT), a regulatory molecule with various biological functions, has been detected in regards to the regulation of seed germination, root development, leaf senescence, and fruit maturation in plants (*Ahammed et al., 2020b*; *Liang et al., 2018a*; *Liang et al., 2018b*). In addition to its physiological functions, MT can also enhance plant tolerance against multiple adverse environmental stressors (*Hardeland et al., 2011*; *Posmyk et al., 2009*; *Li et al., 2018*; *Ahammed et al., 2020b*; *Li et al., 2020*). Numerous studies have shown that MT plays an important role in seed germination and plant growth may be associated with MT-induced changes in physiological mechanisms (*Posmyk et al., 2009*; *Zhao et al., 2011*; *Kang et al., 2010*). Evidence has suggested that the pre-treatment of seeds with MT could improve the effectiveness of pores under drought stress (*Khan et al., 2019*). A previous study conducted on cucumber (*Cucumis sativus* L.) illustrated the positive influences of MT treatment in plants suffering from PEG stress including the promotion of seed germination (*Zhang et al., 2012*). As a broad-spectrum antioxidant, MT can enhance the activity of antioxidant enzymes, such as SOD and POD, by lowering ROS levels, which protects the plants from stress-induced damage (*Gao et al., 2018*; *Wang, Reiter & Chan, 2018*). Previous results have shown that the appropriate concentration of MT can reduce the malondialdehyde (MDA) content in soybean seedlings, alleviating the damage to the membrane system and improving resistance to abiotic stress (*Wei et al., 2014*). Treatment with exogenous MT significantly increased the content of proline, soluble sugar, and soluble protein in plants, while increasing cell fluid concentrations and reducing the MDA content in plants under abiotic stress (*Zhang et al., 2012*; *Turk et al., 2014*). Similarly, the exogenous application of MT has been shown to enhance the activity of α-amylase in maize seeds under chilling stress (*Cao et al., 2019*). MT may be a plant growth hormone that is closely related to auxin (IAA) in its structural components (*Arnao & Hernandez-Ruiz, 2017*). Phytohormones are involved in seed germination and dormancy, and the exogenous application of MT can alleviate damage to seed germination caused by environmental stress by regulating the synthesis of $GA_4$ and the decomposition of ABA (*Zhang et al., 2014a*; *Zhang et al., 2014b*).

Despite the several previously proposed hypotheses suggesting that the exogenous application of MT may promote seed germination under drought stress (*Meng et al., 2014*; *Zhang et al., 2012*), a description of the morphological and physiological mechanisms by which MT plays a role in alleviating drought stress is limited. Furthermore, little is known about the effects of MT on the germination of cotton seeds and its physiology and epidermal microstructure under drought stress. The objectives of our study were to (1) investigate the morphological and physiological changes in cotton seeds induced by soaking the seed in MT and then treating with drought stress and (2) examine the effects of soaking the seed in MT on improving the germination of cotton seeds grown under drought stress.

## MATERIALS & METHODS

### Reagent, plant material, and experimental conditions

MT (N-acetyl-5-methoxytryptamine) was purchased from Sigma-Aldrich (St. Louis, MO, USA). The experiment was conducted in the Key Laboratory of Crop Growth Regulation of

Hebei Agricultural University, Hebei Province, China. Seeds of the widely cultivated cotton cultivar, Guoxin NO. 9 were obtained from Guoxin Agricultural Research Association of Hejian, Hebei Province, China.

## Determination of cotton drought resistance

Seeds of cotton were surface sterilized with 75% ethanol for 30 min and then rinsed with distilled water five times. Sterilized seeds were placed in Petri dishes (each with 50 seeds and five replicates) containing three layers of filter paper and 10 mL of the following treatment solutions: 5, 10, 15, or 20% PEG. Distilled water was used as the control. Then, seeds in Petri dishes were left to germinate in an incubator at $25 \pm 1\,°C$ in the dark. To determine the suitable concentration of PEG-6000, the seed germination rate (GR) was measured at 7 days.

## Determination of melatonin concentration

Selected sterile seeds were soaked in different concentrations of MT solutions (10, 50, 100, 200, and 500 µM MT) for 24 h at 25 °C in a darkened incubator. For the control, the sterilized seeds were soaked in distilled water. Then, 50 seeds were transferred to Petri dishes containing three layers of filter paper moistened with 10 mL of PEG-6000 solution to simulate drought stress and incubated at 25 °C in a darkened growth chamber for 7 days. The selected 10% PEG-6000 concentration was established based on the trial described above. The treatments were replicated five times and each replicate contained 50 seeds. The number of germinated seeds was observed daily and the GR, germination potential (GP), radical length (RL), and fresh weight (FW) were determined to select the effective MT concentration for cotton seed germination under drought stress.

The seeds were pre-soaked with 100 µM MT for 24 h at 25 °C in a darkened incubator and then allowed to germinate following the same methods as described above. Distilled water was used as the control. Approximately 10 g of embryo and radicle were harvested from each treatment at 2, 4, and 6 days after germination, and then stored at $-80\,°C$ for subsequent analyses of SOD and POD activities, osmoregulation, ROS, and hormone content. Four treatments were applied as follows: [1] water soaking + no stress (W); [2] water soaking + 10% PEG-6000 (W+DS); [3] 100 µM MT soaking + no stress (MT); [4] 100 µM MT soaking + 10% PEG-6000 (MT+DS).

Seed coat microstructures were observed in dry, water-soaked, and 100µM MT-soaked cotton seeds for 24 hours at 25 °C using a scanning electron microscope (SEM).

## Assessment of seed germination

The number of seeds germinated was recorded daily. The length of the radicle and hypocotyl was measured with a straightedge ruler. Seeds were considered germinated when their radicle and hypocotyl length exceeded half of the seed length.

The numbers of germinated seeds were calculated as GP and GR at 3 and 7 days after germination, respectively. GP= germinated seeds on day 3 / total seeds × 100%; GR = germinated seeds on day 7 / total seeds × 100% (*Li et al., 2007*; *Thabet et al., 2018*). Germination index (GI) and seed vigor index (VI) were calculated as follows: GI = $\sum$ (Gi / i), where Gi is the number of germinated seeds on day i. VI = GI × RL of germinated

seeds on day 7 (*Li, Yu & Yin, 2017*). In addition, the RL and FW of seeds were determined on day 7 after germination. The weight of 50 seeds was recorded using an analytical balance (0.01 g).

## Measurement of SOD and POD activities and the content of MDA and ROS

The activities of SOD and POD were estimated using the assay kits (A001-1 and A084-3, Nanjing Jiancheng Bioengineering Institute, Nanjing, China). Crushed frozen samples (0.3 g) were homogenized with three mL 50 mM phosphate buffer (PH 7.8) containing 1% polyvinylpyrrolidone and 0.1% mercaptoethanol (w/v), followed by centrifugation at 4 °C for 20 min at 10,000 rpm. The SOD and POD activity was determined in the supernatant using a spectrophotometer at 560 nm and 470 nm based on the manufacturer's instructions, respectively. The activity of SOD and POD was presented as protein content.

The MDA content was tested using the Malondialdehyde Assay Kit (Nanjing Jiancheng Bioengineering Institute, China). Cotton seed tissues were fully homogenized with pH 7.8 phosphate buffer, after centrifugation at 6,000 rpm for 10 min. Two milliliters of supernatant were added to a test tube contained three mL of 5% trichloroacetic acid and one mL of 0.5% thiobarbituric acid solution. Then the mixture was boiled for 10 min, cooled rapidly, and centrifuged at 6,000 rpm for 10 min. The light absorption value was measured in the resulting supernatant using a spectrophotometer at 532 nm and 600 nm.

The $H_2O_2$ and $O_2^-$ content were tested using a Hydrogen Peroxide Assay Kit and Superoxide Anion Assay Kit (Nanjing Jiancheng Bioengineering Institute, Nanjing, China) following the manufacturer's protocols.

## Determination of soluble sugar, soluble protein, proline, and amylase contents

The method described by *Zhang et al. (2006)* was adapted for the estimation of soluble sugar contents with some slight modification. Briefly, 0.3 g seed samples were added to nine mL of distilled water in a boiling water bath for 30 min. One milliliter of the supernatant was mixed with five mL of sulfuric acid-anthrone reagent, followed by boiling for 10 min and cooling. The absorption value was measured in a spectrophotometer at 620 nm.

The proline content was estimated following the method described by *Subramanyam, Du & Van (2019)*. Briefly, approximately 0.3 g frozen samples were homogenized with three mL of 3% aqueous sulfosalicylic acid. After centrifugation, two mL of supernatant was mixed with two mL of glacial acetic acid and two mL of ninhydrin reagent, and then heated in boiling water for 30 min. After cooling, the mixture was centrifuged at 10,000 rpm for 5 min. The light absorption values were recorded at 520 nm using a spectrophotometer.

The soluble protein content was determined according to the Coomassie brilliant blue (CBB) method described by *Yasmeen et al. (2013)*. The α-amylase content was calculated according to the α-Amylase Assay Kit (Nanjing Jiancheng Bioengineering Institute, Nanjing, China).

## Quantification of GA$_3$ and ABA

GA$_3$ and ABA concentrations were determined using a previously described indirect ELISA technique (*He, 1993*). To assay GA$_3$ and ABA, 0.3 g frozen tissue were ground in liquid nitrogen. After homogenization, four mL of 80% methanol was added to the samples. The solution was then incubated for 4 h at 4 °C in the dark. After centrifugation at 10,000 g for 20 min, the supernatant was eluted using a Sep-Pak C18 cartridge and then dried under a stream of N$_2$. The dried samples were re-dissolved with five mL of eluting buffer containing 1 mM MgCl$_2$, 150 mM NaCl, and 10% (v/v) methanol in 50 mM Tris. GA$_3$ and ABA were determined with an immunosorbent assay according to the manufacturer's instructions, and the antibodies used in this assay were all monoclonal antibodies provided by China Agricultural University.

## Scanning electron microscopy (SEM) analysis

Parts of the testa were collected from healthy cotton seeds and observed using SEM (Hitachi SU8020, Japan). The collected samples were fixed using 3% glutaraldehyde for 12 h at 4 °C, and then washed using 0.1 M sodium phosphate buffer. Samples were fixed with 1% osmium for 1 h at 4 °C and washed twice with 0.1 M sodium phosphate buffer. Samples were dehydrated in serially diluted ethanol and critical point dried using carbon dioxide. The samples were firmly mounted using double-sided adhesive tape on stubs and then sprayed with gold using a vacuum plating instrument.

## Statistical analysis

We conducted five independent replicates for germination tests. Physiological parameters were assessed using three replicates. The data were statistically analyzed using IBM SPSS Statistics 17.0 software. A least significant difference test was determined using Duncan's test to compare the mean value. All data were expressed as mean ± standard deviation (SD). Differences were considered significant at $P < 0.05$. The figures were developed using the GraphPad Prism 8.0 software.

# RESULTS

## Selection of PEG concentration

Cotton seeds treated with different concentrations of PEG-6000 (0–20%) were used to determine the optimum concentration of PEG-6000. Figure 1 illustrates the daily germination rate of cotton seeds treated with varying concentrations of PEG-6000 within 7 days of seed germination. Figure 2 shows the effects of different degrees of drought stress on the seed germination rate. The increase in PEG concentration resulted in a significant delay in the germination rate of seeds. Treatment with 5, 10, 15, and 20% PEG-6000 significantly reduced the germination rate by 3.36%, 20.17%, 37.82%, and 57.14%, respectively, compared with the control treatment. The inhibitory effect of 5% PEG on seed germination was weak, whereas the degree of stress of 15% and 20% PEG severely inhibited seed germination; treatment with 10% PEG created a moderate level of stress on seed germination. Therefore, in subsequent experiments, 10% PEG was selected for the drought stress treatment.

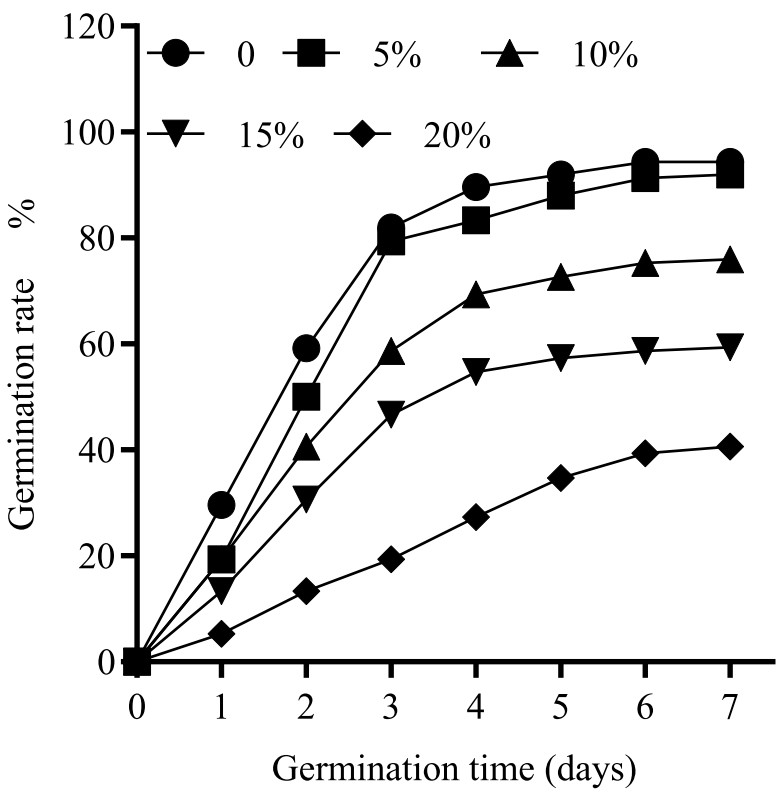

**Figure 1** The daily germination rate of seeds treated with different concentrations of PEG-6000.

## Selection of MT concentrations for seed treatment

We conducted seed germination tests to determine whether different concentrations of MT affect seed germination under drought stress. In addition, we measured the radical length and fresh weight of the seeds 7 days after germination. As shown in Table 1, treatment with different concentrations of MT limited the negative effects of PEG-induced drought stress on germination. Further analysis showed that a low concentration of MT (10–50 μM) had a limited effect on GP, GR, RL, and FW under drought stress conditions. Pre-soaking seeds with 100 μM MT was the most effective at relieving drought stress, and the results showed that GP, GR, RL, and FW increased by 8.99%, 10.42%, 7.62%, and 53.37% respectively, compared to the 0 μM MT treatment. Fewer seeds germinated when the seeds were pre-soaked in 200 μM or 500 μM MT. Overall, the results showed that pre-treatment with MT at an appropriate concentration could promote seed germination under drought stress. Based on these results, we selected 100 μM MT for subsequent germination tests.

## Effect of MT on germination indices under drought stress conditions

The seeds pre-soaked in100 μM MT showed no significant change for germination indices under the well-watered condition (Table 2). MT+DS treatment resulted in a significantly higher GP (8.9%) and GR (9.8%) when the seeds were germinated under drought stress conditions compared with the W+DS treatment. At 10% PEG-6000, MT acted as a good regulator, significantly improving GI and VI under drought stress. Further analysis showed
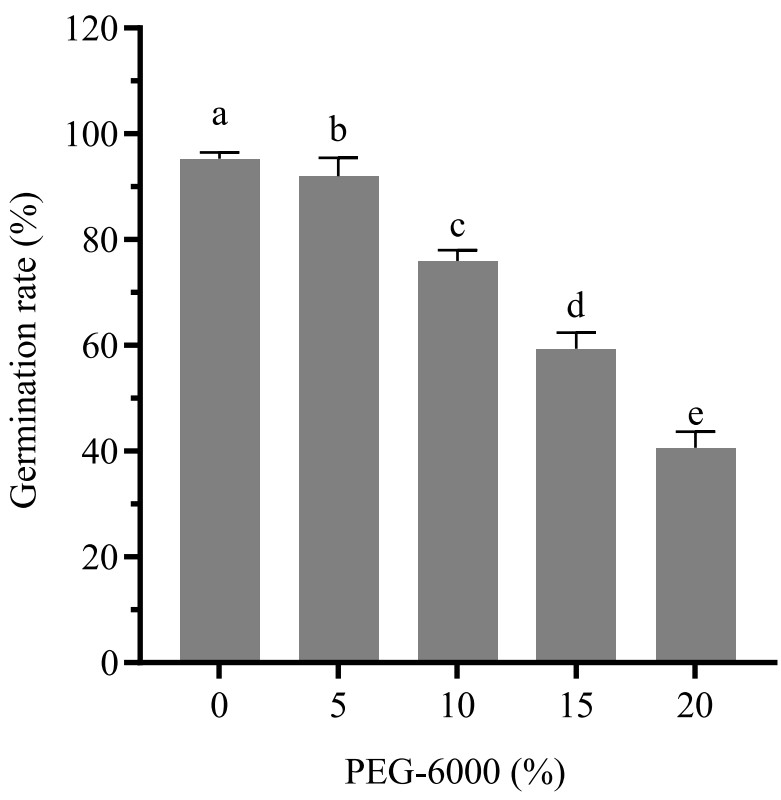

**Figure 2** **Effects of drought stress on the germination rate of cotton seeds.**

**Table 1** **Effects of different concentrations of melatonin on the germination and growth of seeds under drought stress.** Cotton seeds were pre-soaked in MT solutions of 0 (control), 10, 50, 100, 200, and 500 µM for 24 h and germination under 10% PEG stress. Different lowercase letters indicate a significant difference at the 0.05 probability level ($P < 0.05$). Error bars indicate standard errors calculated for five replications.

| Melatonin concentration (µM) | GP (%) | GR (%) | RL (cm) | FW (g) |
|---|---|---|---|---|
| 0 | $44.5 \pm 1$ b | $72 \pm 3.65$ c | $6.3 \pm 0.16$ b | $11.49 \pm 1.06$ c |
| 10 | $43.2 \pm 2.68$ b | $76 \pm 1.41$ b | $6.15 \pm 0.11$b | $13.84 \pm 0.89$ b |
| 50 | $43.2 \pm 2.28$ b | $73.5 \pm 1.91$ bc | $6.6 \pm 0.08$ a | $13.32 \pm 0.69$ b |
| 100 | $48.5 \pm 1.91$ a | $79.5 \pm 1.91$ a | $6.78 \pm 0.19$ a | $17.63 \pm 0.94$ a |
| 200 | $33.2 \pm 2.28$ c | $63.2 \pm 1.09$ d | $5 \pm 0.12$ c | $14.21 \pm 0.98$ b |
| 500 | $28.4 \pm 2.61$ d | $50.5 \pm 1$ e | $2.46 \pm 0.36$ d | $13.72 \pm 0.88$ b |

that the treatment of MT+DS increased GI and VI by 13.4% and 16.2%, respectively, compared with W+DS. Therefore, pre-treatment with MT could positively affect seed vigor.

**Table 2  Effect of melatonin treatment on GP, GR, GI, and VI of cotton under drought stress.** Cotton seeds were pre-soaked in the solutions of 0 (control) and 100 μM MT for 24 h and germination under 0 (control) and 10% PEG stress. Different lowercase letters indicate a significant difference at the 0.05 probability level ($P < 0.05$). Error bars indicate standard errors calculated for five replicates.

| Treatments | GP (%) | GR (%) | GI (%) | VI (%) |
|---|---|---|---|---|
| W | 51.5 ± 1 a | 90 ± 1 a | 18.29 ± 1.11 a | 137.55 ± 5.29 a |
| W+DS | 44.5 ± 1 c | 72.25 ± 4.03 c | 15.37 ± 0.64c | 103.44 ± 1.83 c |
| MT | 52.67 ± 0.58 a | 87.67 ± 1.53 a | 19.36 ± 0.98 a | 121.97 ± 6.16 b |
| MT+DS | 48.51 ± 0.57 b | 79.33 ± 2.31 b | 17.44 ± 0.61 b | 120.15 ± 4.31 b |

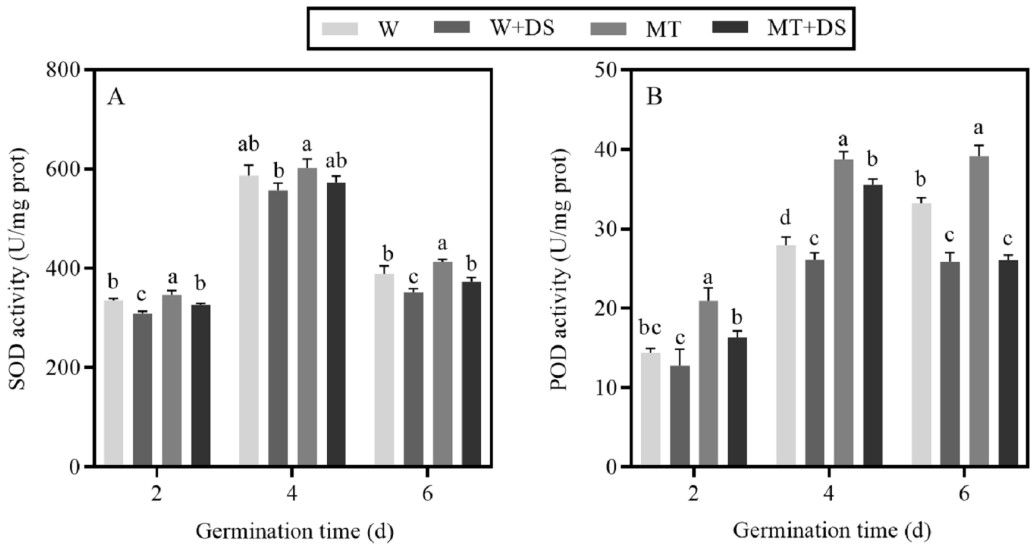

**Figure 3  Effect of melatonin treatment on (A) superoxide dismutase (SOD) and (B) peroxidase (POD) activities of cotton seeds under drought stress.**

## Effects of melatonin on SOD and POD activities of cotton under drought stress

We determined the changes of SOD and POD activities at different stages of germination. Figure 3A shows that the SOD activity of each treatment increased first and then decreased, and the highest SOD activity was found 4 days after germination. Two days after seed germination under well-watered conditions, the activity of SOD significantly increased by 3.3% in MT compared with that in W. In the seeds pre-soaked with MT, when cotton seeds were grown in normal moisture conditions for 4 and 6 days, the SOD activity increased by 2.6% and 6.3%, respectively, compared to the W treatment, but there was no statistical difference 4 days after germination. Similarly, the activity of SOD was enhanced by the pre-treatment of seeds with MT when the seeds were germinated under drought stress conditions. Compared with the W+DS treatment, the SOD activity of the MT+DS

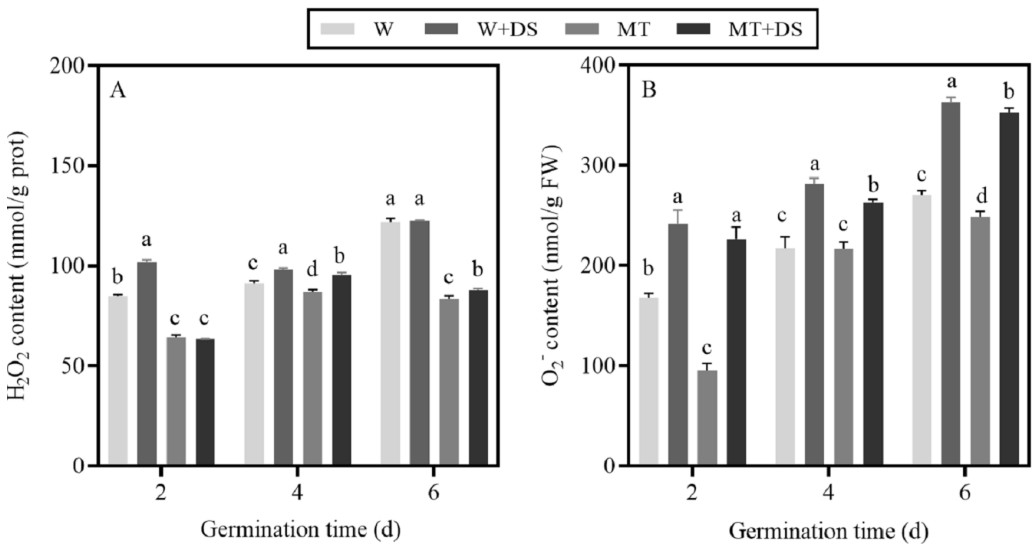

**Figure 4** **Effects of pre-treatment with melatonin on $H_2O_2$ (A) and $O_2^-$ (B) content of cotton grown under drought stress conditions.**

treatment significantly increased by 5.69%, 2.91%, and 6.03% at days 2, 4, and 6 after germination, respectively.

The results of the POD activity analysis were consistent with that of the SOD activity data (Fig. 3B). In all cases, POD activity was substantially improved by pre-soaking seeds with MT compared to seeds pre-treated with distilled water. Under well-watered conditions, POD activity was increased 1.5 times by seeds pre-soaked with MT compared with the W treatment at 2 days after germination. Compared with W, the POD activity of the MT treatment improved by 38.74% and 17.9% at 4 and 6 days after germination, respectively. Soaking seeds in MT enhanced seed vigor; therefore, the POD activity of MT+DS on days 2 and 4 of germination under drought stress significantly increased by 28.28% and 36.23% respectively, compared with the W+DS treatment. Six days after seed germination, the POD activity in the MT+DS treatment increased by 0.6%, but this increase was not statistically significant.

## Effects of melatonin pre-treatment on ROS in seeds under drought stress conditions

The accumulation of ROS significantly increased under drought stress. On the contrary, pre-soaking seeds with MT resulted in relatively low $H_2O_2$ and $O_2^-$ content (Fig. 4). The $H_2O_2$ content in the seeds pre-soaked with MT under no stress conditions significantly decreased by 24.4%, 4.3%, and 31.6%, respectively at 2, 4, and 6 days, compared with W treatment. Under drought stress conditions, the MT+DS treatment caused a decrease by 37.91%, 2.85%, and 30.31% at 2, 4, and 6 days after germination, respectively compared with the W+DS treatment (Fig. 4A).

The $O_2^-$ content showed an upward trend over the course of seed germination. Under normal growth conditions, the $O_2^-$ content increased in the W treatment. However, the

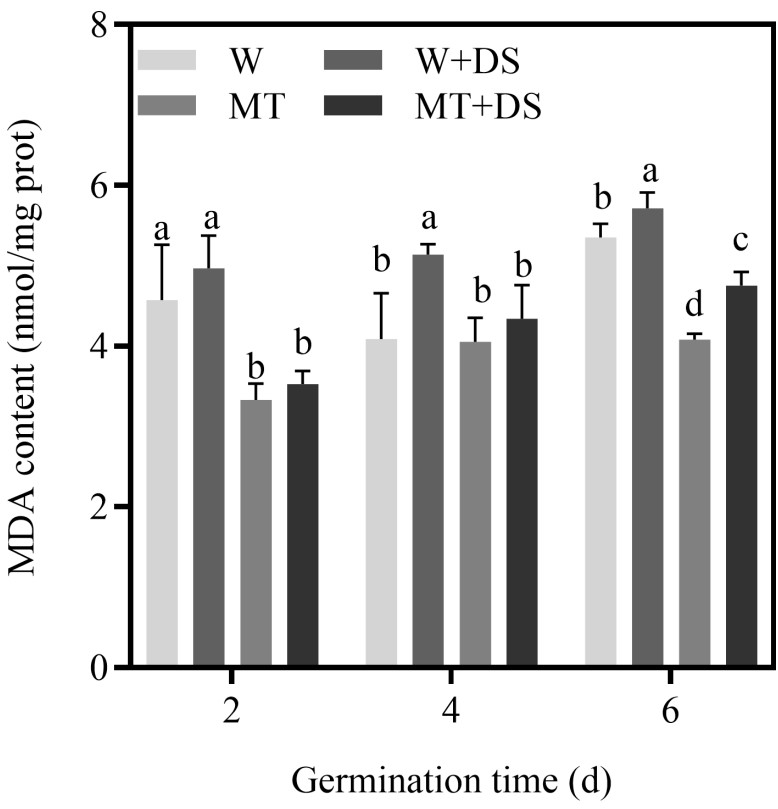

**Figure 5** Effects of melatonin treatment on MDA content of cotton under drought stress.

increased $O_2^-$ was alleviated by pre-soaking seeds with MT. On 2, 4, and 6 days after seed germination, the content of $O_2^-$ in the MT treatment decreased by 43.1%, 0.3%, and 8.1%, respectively, compared with W. A similar effect was observed under drought stress conditions at 2, 4, and 6 days after germination; specifically, the MT+DS treatment reduced $O_2^-$ content by 6.16%, 6.81%, and 2.82%, respectively, compared to the W+DS treatment. These data indicate that MT can regulate ROS during germination in the drought stress treatment (Fig. 4B).

## Effects of melatonin treatment on the MDA content of cotton under drought stress

MDA is an important indicator reflecting the degree of membrane lipid peroxidation. The data showed that drought stress induces the increase in MDA content, which was mitigated by pre-soaking seeds with MT. Under drought stress, the MT+DS treatment resulted in 29.07%, 15.58%, and 16.83% lower MDA content compared with the W+DS treatment at 2, 4, and 6 days after seed germination, respectively, which further suggests the effects of MT as an antioxidant (Fig. 5).

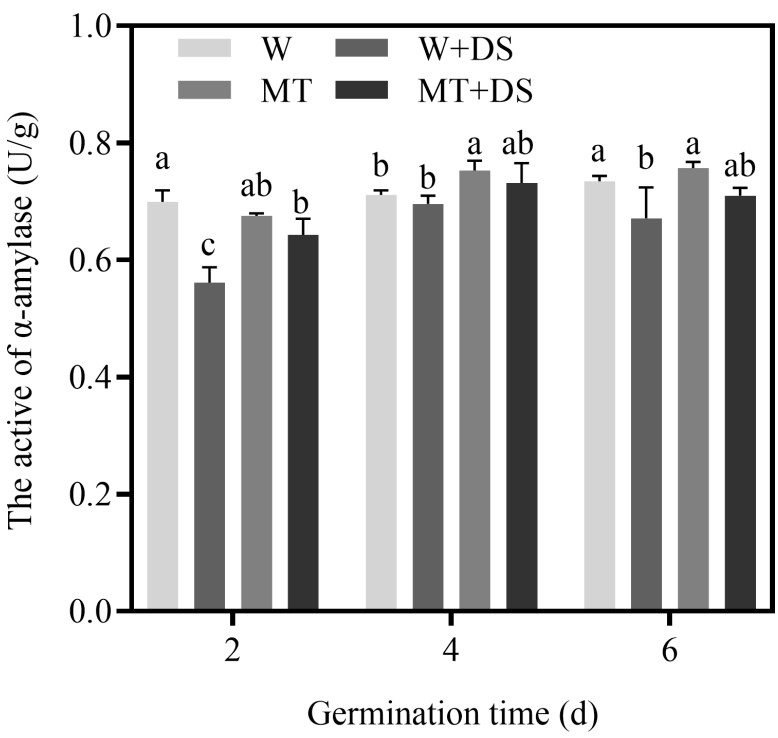

**Figure 6 Effects of melatonin treatment on α-amylase activity of cotton under drought stress.**

## Effect of melatonin treatment on amylase activity of seeds under drought stress

In our experiment, α-amylase activity was analyzed during seed germination. The activity of α-amylase under drought stress was decreased (Fig. 6). In contrast, seed soaking with 100 μM MT significantly enhanced the activities of α-amylase. Two days after germination, the activity of α-amylase on the MT+DS treatment significantly increased by 14.5% compared with W+DS under drought stress. MT showed higher α-amylase activity compared with W and was increased by 5.9% and 3.5% at 4 and 6 days after germination under normal conditions, respectively (Fig. 6). Similarly, under drought stress conditions, the α-amylase activity in the MT+DS treatment increased by 5.2% and 5.8% on the fourth and sixth day after germination, respectively, compared to the W+DS treatment. Therefore, under drought stress conditions, the α-amylase activity was effectively improved in seeds pre-soaked with 100 μM MT.

## Effects of melatonin treatment on osmoregulatory substances in seeds under drought stress

As shown in Fig. 7A, drought stress enhanced the accumulation of proline content in seeds. Under well-watered conditions, seeds pre-soaked with MT displayed a negligible effect on proline content. While the MT+DS treatment significantly improved the proline content compared with the W+DS treatment under drought stress. Our data showed that the proline content of the MT+DS treatment significantly increased by 7.11%, 9.92%, and

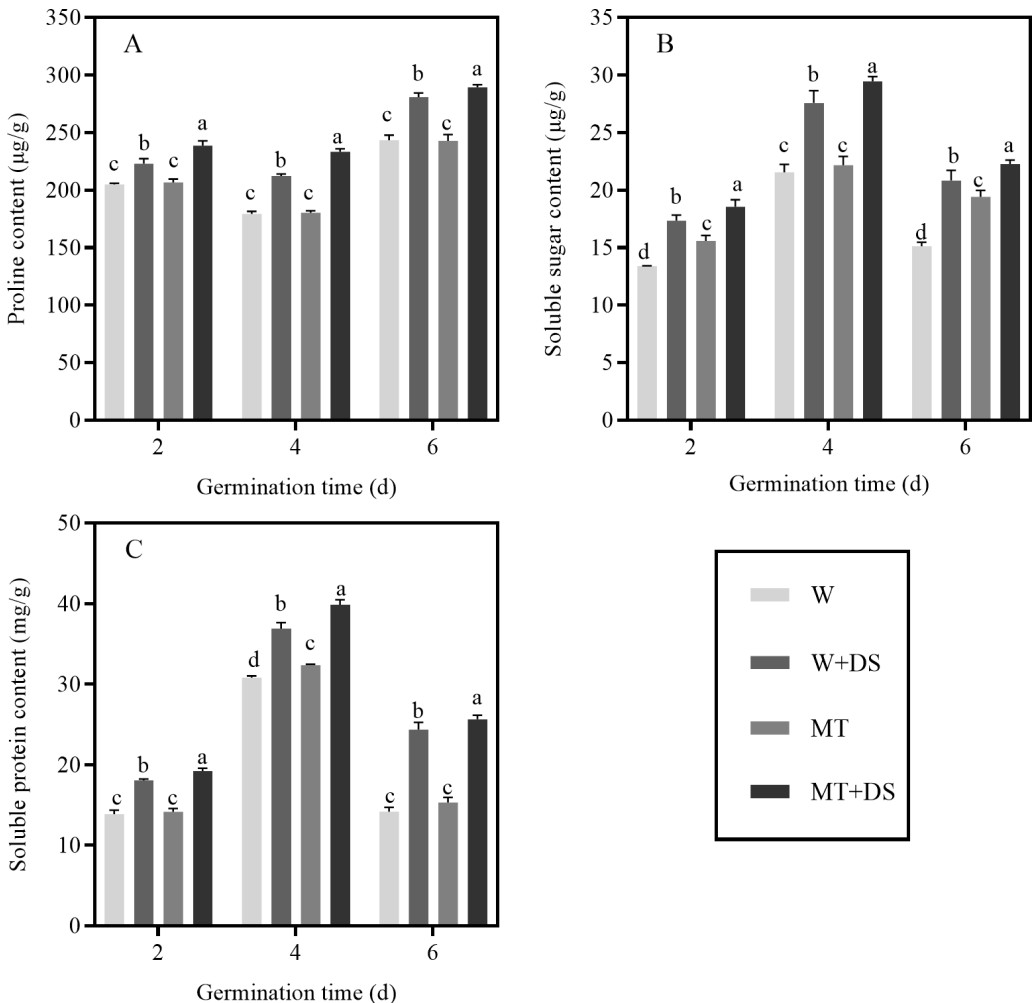

**Figure 7** **Effects of melatonin treatment on proline (A), soluble sugar (B), and soluble protein (C) content of cotton seeds under drought stress.**

3.08% compared with the W+DS treatment at 2, 4, and 6 days after seed germination, respectively.

Drought stress also increased the soluble sugar content. The soluble sugar content in seeds pre-soaked with MT was higher than seeds not pre-treated with MT. Under normal conditions, the soluble sugar content increased by 16.37%, 2.85%, and 28.22% in the MT treatment at 2, 4, and 6 days after germination, respectively, compared with W treatment. Under drought stress, the soluble sugar content in the MT+DS treatment was 6.91%, 6.82%, and 6.89% higher at 2, 4, and 6 days after germination, respectively, compared with the W+DS treatment (Fig. 7B).

The soluble protein content also responded positively to pre-soaking seeds with MT. Pre-treatment with MT considerably inhibited the decrease of soluble protein content under all conditions. The soluble protein content under the MT treatment was increased by 2.16%, 5.03%, and 7.98% at 2, 4, and 6 days after seed germination, respectively, compared

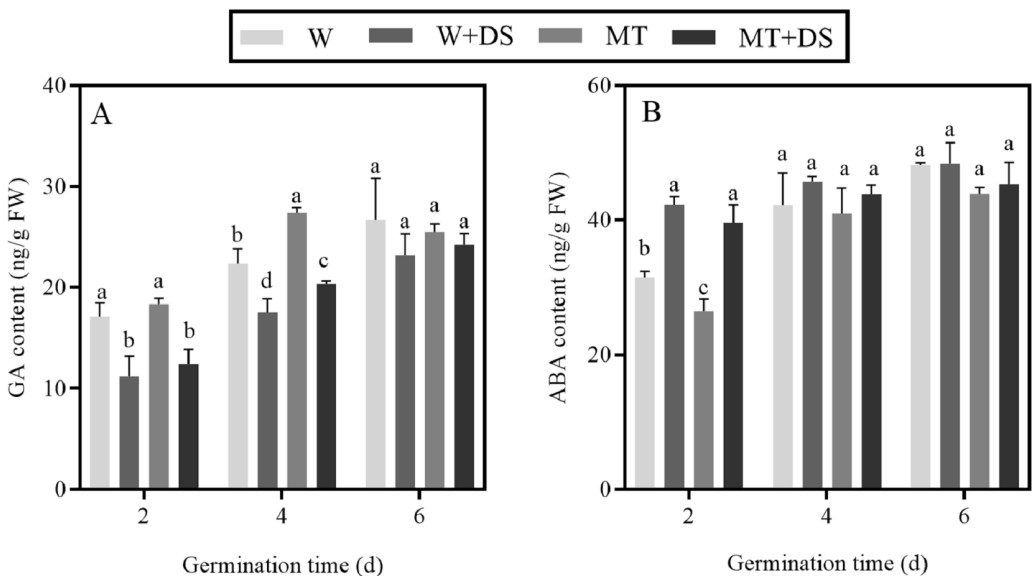

**Figure 8** Effects of melatonin treatment on contents of gibberellin (GA) (A) and abscisic acid (ABA) (B) under drought stress.

to the W treatment. Under drought stress conditions, the soluble protein content in the MT+DS treatment increased by 6.36% and 7.95%, respectively, at 2 and 4 days after seed germination compared with the W+DS treatment. However, 6 days after germination, the effect of the MT treatment on the protein content was more effective under drought stress than under the water treatment; the protein content in the MT+DS treated cotton seeds increased by 5.22% compared with the W+DS treatment. Consequently, we can infer that MT may accelerate the germination of cotton seeds under drought stress conditions (Fig. 7C).

## Effects of the melatonin treatment on endogenous hormones under drought stress

As shown in Fig. 8A, GA$_3$ continuously increased over the course of germination; however, drought stress conditions negatively affected GA$_3$ biosynthesis. The GA$_3$ content was greater in the seeds pre-soaked with MT compared with those not pre-soaked with MT under drought stress conditions. The GA$_3$ content increased by 10.81% in the MT+DS treatment, compared with the W+DS treatment 2 days after germination under drought stress conditions; however, this difference was not significant. Four days after germination, the GA$_3$ accumulation significantly increased by 16.09% in the MT+DS treatment compared with the W+DS. Thus, this treatment alleviated the inhibitory effects of drought stress. At the end of the germination experiment, pre-treatment with MT further promoted GA$_3$ synthesis in seeds, but there was no significant difference compared with the W+DS

treatment. The effect of MT on drought stress during germination could be via enhancing $GA_3$ biosynthesis.

Similar to $GA_3$, the ABA content increased over the course of germination (Fig. 8B). In this study, MT accelerated ABA catabolism during germination in all cases. In response to drought induced by treatment with PEG-6000, ABA catabolism was limited during seed germination. Additionally, the ABA content in the seeds pre-soaked with MT was lower compared to the W treatment under normal conditions. Under drought stress conditions, the ABA content of the MT+DS treatment decreased by 6.22%, 4.2%, and 6.42% 2, 4, and 6 days after germination, respectively, compared with the W+DS. These results indicated that the mechanism by which MT reduced PEG-6000 stress was by promoting the degradation of ABA.

## Microstructural observation of testa

Figure 9A shows that the seed coat stomata were rare on the surface of dry seeds. Only a small number of pores were observed in seeds soaked in water (Fig. 9B). However, seeds soaked in MT showed a greater number of pores (Fig. 9C). The stomata were located on the surface of the epidermis between the ridges of the irregular epidermal cells, but the inner side of the stomata was not surrounded by epidermis. The pores were oval and not indented on the surface of the dry seeds (Fig. 9D). The periphery of the stomata was elliptic and ranged from oval to round toward the center, but the inner guard cells were covered with cuticular remnants in seeds soaked in water (Fig. 9E). However, soaking seeds with MT played a significant role in the opening of pores in the testa. Compared with dry seeds and seeds soaked in water, the pores of seeds soaked in MT were completely open and contained large guard cells (Fig. 9F).

## DISCUSSION

Drought stress markedly impairs plant growth via different mechanisms and influences seed germination and metabolism (*Zhang et al., 2012*). The immediate response of a plant to drought stress can be determined by the inhibition of the germination rate. The damage caused by drought stress to seed germination is mainly due to decrease in water intake and reduced energy supply to seeds, inducing a series of changes in metabolism, including ROS accumulation, antioxidant defense system, hormone signal transduction, and the production of osmotic regulatory substances (*Hussain, Farooq & Lee, 2017*; *Shu et al., 2018*; *Sharma & Zheng, 2019*). Numerous studies have demonstrated that drought stress significantly inhibits seed germination in various plant species (*Liu et al., 2018*; *Lou et al., 2017*; *Liu et al., 2016a*; *Liu et al., 2016b*; *Liu et al., 2019*). Our experiments showed complementary evidence that the germination rate of cotton seeds was inhibited by the application of 10% PEG-6000 (Fig. 2) because it caused a negative regulation in physiological mechanisms and inhibited water absorption by seeds, suggesting that drought stress affected cotton seed germination.

Recently, plant growth regulators have been widely used to treat seeds before sowing, and can improve their resistance to abiotic stressors and promote seed germination (*Liang*

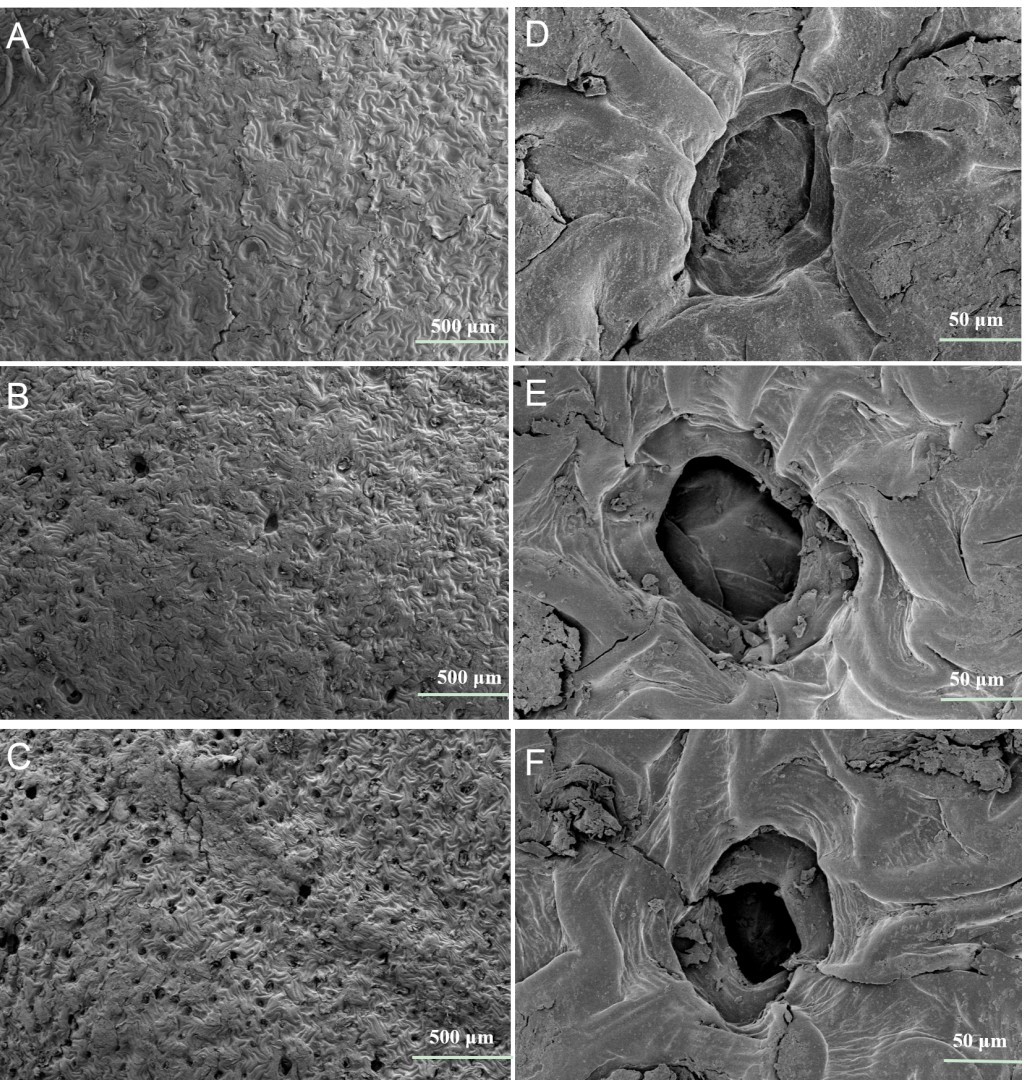

**Figure 9  Scanning electron microscopic of the testa of cotton seeds.** (A and D) represent the dry seed; (B and E) represent the seed soaked in water for 24 h at 25 °C; (C and F) represent the seed soaked in MT solution for 24 h at 25 °C (A–C, scale bars = 500 μm; D–F, scale bars = 50 μm).

*et al., 2018a*; *Liang et al., 2018b*; *Zheng et al., 2016*). In the present study, seed germination inhibition caused by PEG-6000 was alleviated by soaking cotton seeds with MT. Under the same drought stress treatment, seeds soaked with 100 μM MT showed an accelerated seed germination rate and improved radical length and fresh weight compared with seeds not pre-treated with MT. In addition, 10–50 μM MT treatments had little effect on cotton seed germination. However, 200–500 μM MT exerted a significantly negative effect on cotton seed germination (Table 1). These results demonstrated that the effect of MT on cotton seeds is closely associated with its concentration and 100 μM MT was defined as the optimum concentration to promote cotton seed germination under drought stress. These findings are in accordance with *Simlat et al. (2018)*, who demonstrated that a low

concentration of exogenous MT promoted seed germination, while treatments with a high dose of MT inhibited seed germination. Previous studies reported that soaking seeds in exogenous MT (20 and 100 µM) alleviated drought-induced germination inhibition by advancing the antioxidant defense system (*Ding, Liu & Zhang, 2017*). The germination test indicated that treatment with 100 µM MT (MT+DS) increased the GP, GR, GI, and VI compared with the seeds only pre-soaked with water (W+DS) under drought stress conditions (Table 2). Our findings are consistent with previous results showing that the application of exogenous MT could alleviate the inhibition of PEG stress on cucumber seed germination (*Zhang et al., 2014a*; *Zhang et al., 2014b*).

The decisive factors for seed germination are water and oxygen, which are absorbed and transmitted through the seed coat (*Léonie & Maarten, 2008*; *Kebede, Smith & Ray, 2014*). The increase in seed coat stomata promotes the intake of water and oxygen from the environment, resulting in seed germination (*López-Ribera & Carlos, 2017*; *Wang & Hasenstein, 2016*); therefore, the seed coat stomata play a vital role in seed germination and growth. The stomata consist of two guard cells and subsidiary cells (*Wang & Hasenstein, 2016*). Previous studies have shown that the absorption rate of water is highly related to pore density (*Paiva, Lemos & Oliveira, 2006*). The number of seed stomata is not necessarily species-specific and may depend on environmental conditions (*Wang & Hasenstein, 2016*). In the current study, few stomata were observed in dry seeds, suggesting that the lack of stomata may be due to dehydration in maturing seeds. The variability in the number and shapes of pores indicates that the number of stomata is fluid and may have some physiological function (*Wang & Hasenstein, 2016*). The results of the SEM showed that pre-treatment with MT increased the number of stomata and accelerated the opening of pores in testa, confirming that exogenous MT stimulated the physiological function of stomata and accelerated the water absorption rate of seeds (Fig. 9). Therefore, a vital effect of the MT treatment is the induction of the opening of stomata in the cotton seed coat. While a quantitative analysis of stomata in testa was not carried out, our results do suggest that the function of MT on the regulation mechanism of seed coat stomata requires further study.

Seed germination is controlled by physiological factors and ROS concentration is closely correlated with water uptake by seeds during the process of germination (*Moothoopadayachie et al., 2016*). The balance between the production and elimination of ROS is disturbed by drought stress, which leads to the accumulation of ROS, including $O_2^-$ and $H_2O_2$, and leads to the instability of cell membranes (*Kaur et al., 2015*; *Yang, Yao & Zhang, 2010*; *Li et al., 2015*; *Sohag et al., 2020*). $H_2O_2$ content is associated with embryo elongation during seed germination (*Zhang et al., 2014a*; *Zhang et al., 2014b*). MT is a free radical scavenger, which may directly or indirectly eliminate ROS (*Reiter et al., 2007*; *Tan et al., 2000*). Previous research has suggested that pre-treatment with MT effectively inhibited the accumulation of $H_2O_2$ in corn seeds under stress conditions (*Cao et al., 2019*). Similarly, the application of MT reduced the accumulation of ROS and alleviated membrane damage of tomato seeds and cucumber seedlings under the drought stress (*Zhang et al., 2012*; *Liu et al., 2015*). It is necessary to determine whether MT can promote the germination of cotton seeds by inhibiting the accumulation of ROS. We found that seeds pre-soaked in distilled

water and treated with normal and dry conditions had an increase in $H_2O_2$ and $O_2^-$ contents; however, the $O_2^-$ and $H_2O_2$ contents were decreased in seeds pre-soaked with MT (Fig. 4). This may be because MT was instrumental in accelerating water absorption by the seed and maintaining the balance between ROS production and clearance. Furthermore, this may have led to repair of the cell membrane even under drought stress conditions. These results provided evidence that exogenous MT played an important role in removing excessive ROS and alleviating the membrane lipid peroxidation in cotton seeds faced with drought stress.

To inhibit excessive generation of ROS, plants possess a complex enzymatic defense system to combat oxidative damage (Souza et al., 2014; Ahammed et al., 2020c). SOD and POD act as important antioxidant enzymes that can eliminate redundant ROS from plant tissues, protecting the plasma membrane from peroxidation (Xiao et al., 2019; Ramachandra et al., 2004). In plant cells, SOD can rapidly convert $O_2^-$ to form $H_2O_2$ and $O_2$, while $H_2O_2$ can be further scavenged by other antioxidant enzymes (Li et al., 2017). Many studies have shown that MT plays a vital role in enhancing the activity of antioxidant enzymes and scavenging radicals in response to stress-induced ROS damage (Zhang et al., 2014a; Zhang et al., 2014b; Sharif et al., 2018; Jiang et al., 2016; Shi et al., 2015). In the present study, the activity of SOD and POD were low at 2 d of germination, peaked at 4 d, and gradually decreased by 6 d, indicating that the antioxidant enzyme system was active in the middle of seed germination. Drought stress led to the inhibition of SOD and POD activities in cotton seeds, which may be because drought stress induced the excessive generation of ROS, negatively affecting antioxidant enzyme activities. We treated cotton seeds with exogenous MT to enhance the tolerance of cotton seeds to drought stress. We found that seeds pre-soaked with MT enhanced the activities of the SOD and POD both under normal water and drought stress conditions by inhibiting the accumulation of ROS (Fig. 3). These results were in consistent with a previous study that described that treatment with exogenous MT promoted SOD and POD activities in grape exposed to drought conditions (Meng et al., 2014).

The accumulation of ROS induces lipid peroxidation and the degradation of unsaturated fatty acids, producing MDA, which causes damage to the seed structure and reduces seed germination (Rajjou & Debeaujon, 2008; Xiao et al., 2019). Previous studies have shown that the application of exogenous MT significantly suppressed MDA accumulation under abiotic stressors in several plant species (Ahammed et al., 2019b; Hasan et al., 2019). Zhang et al. (2014a) and Zhang et al. (2014b) demonstrated that treatment with 300 μM MT, which increased the activity of antioxidant enzymes, could effectively reduce the MDA content in cucumber seeds under drought stress. In the present study, the MDA content gradually increased during the seed germination assay. Drought stress increased the MDA content and induced lipid membrane peroxidation. However, pre-soaking seeds with 100 μM MT decreased the accumulation of MDA compared with seeds pre-soaked in water under both normal and drought conditions (Fig. 5), suggesting that MT enhanced the activity of antioxidant enzymes potentially by eliminating ROS and reducing membrane lipid peroxidation in cotton seeds (Ahammed et al., 2020b).

Seed germination is regulated by the presence of several storage products (*Waterworth et al., 2016*; *Fleming, Richards & Walters, 2017*). Amylase and osmotic regulators (e.g., proline, sugars, and protein) are compatible solutes that provide necessary energy for seed morphogenesis. Furthermore, amylase protects plants from stress conditions (*Sadura & Janeczko, 2018*) by accumulating in the form of an inactive polymer and storing in the endosperm of dry seeds as storage protein, which is released and activated gradually during germination; therefore, amylase plays an important role in seed germination (*Swanston & Molina-Cano, 2001*). Pre-treatment with exogenous MT has been revealed to increase α-amylase activity in *Limonium bicolor* seeds and accelerate seed germination under salt stress (*Li et al., 2019*). In the present study, under normal conditions, MT treatment was not positively correlated with α-amylase activity at 2 d of germination. However, the α-amylase activity increased in response to MT treatment by days 4 and 6, confirming that the effectiveness of exogenous MT on amylase activity may not be apparent at the beginning of germination under normal conditions. The MT+DS treatment enhanced the activity of α-amylase at all germination time points under drought stress, compared with the W+DS treatment (Fig. 6), suggesting that MT may promote starch hydrolysis and seed respiration, improving seed germination and growth.

Proline is an important osmotic regulator and free radical scavenger that can alleviate stress damage by reducing water potential (*Hayat et al., 2012*; *Bala, 2000*). We found that the proline content of each treatment increased gradually throughout the process of seed germination. The effect of the MT treatment was similar to that of W treatment and did not show a positive role in increasing proline content. However, the MT+DS treatment performed better under drought stress compared with the W+DS treatment, as evident from the increased proline content (Fig. 7A). These results indicate that MT may regulate the metabolism of osmotic substances under drought stress and improve drought tolerance. Pre-treatment with 100 µM MT can improve the drought resistance of cotton seeds, which is in agreement with a previous study that found that pre-treatment with MT increases the proline content in *Brassica napus* exposed to Se-stress (*Ulhassan et al., 2019*). Soluble sugar and soluble protein, which play a role in osmoregulation and cell metabolism, are the main components of organelles (*Wang et al., 2004*; *Zhang et al., 2015a*). This study found that the content of soluble sugar and protein first increased and then decreased during the process of seed germination. The contents of soluble sugar and protein in the MT+DS treatment were higher than in the W+DS treatment, and the exogenous application of MT may be a particularly important for the accumulation of osmotic regulators under drought stress conditions (Fig. 7). These findings indicate that MT could reduce the damage to plant cells caused by drought stress, reduce excessive water loss, and enhance the drought resistance of plants. These findings are similar to those presented by *Jiang et al. (2016)*.

Plant hormones play vital roles in the life cycle of plants, including seed germination, dormancy, and response to abiotic stress (*Zhou et al., 2019*; *Liu et al., 2016a*; *Liu et al., 2016b*; *Shu et al., 2016*). It is generally recognized that ABA and GA regulate seed germination antagonistically (*Okamoto et al., 2006*); GA promotes seed germination by breaking seed dormancy, while ABA inhibits seed germination (*Zhou et al., 2019*; *Yang et al., 2014*; *Okamoto et al., 2006*; *Toh et al., 2008*). Pre-treatment with MT enhanced the stress

resistance of plants by regulating the changes of endogenous hormones (*Jiang et al., 2016*). *Zhang et al. (2014a)* and *Zhang et al. (2014b)* showed that the exogenous application of MT significantly increased the GA content and accelerated ABA degradation in cucumber seeds under salt stress. Our results showed that ABA content was increased, whereas $GA_3$ content was decreased under drought stress conditions. Compared with the W+DS treatment, the MT+DS treatment increased the $GA_3$ content and inhibited the production of ABA content in cotton seeds grown under drought stress conditions (Fig. 8), which is indicative of the role of MT on regulating the endogenous hormones in cotton under drought stress. Similarly, the pre-treatment of cotton seeds with MT can also promote the accumulation of $GA_3$ and decrease the ABA content under normal water conditions; however, the effect of the MT treatment on the $GA_3$ content was not obvious at day 6. This may be because the effect of MT on $GA_3$ biosynthesis was inconspicuous at the later stage of germination in the normal water treatment. This confirmed that the application of MT can maintain the balance of $GA_3$ and ABA, effectively accelerating the germination of cotton seeds under drought stress.

## CONCLUSIONS

Pre-treatment of cotton seeds with MT improves germination under drought stress conditions. The results of our ultrastructure analysis of seed coats demonstrated that pre-treatment with MT increased the number of stomata and promoted the opening of pores. Likely, this enhanced antioxidant enzyme activities and reduced drought-induced ROS accumulation. Pre-treatment with MT also increased starch metabolism and the content of osmotic substances, overall enhancing the germination rate of cotton seeds. Pre-treatment with 100 μM MT may promote germination by regulating the hormone content; seeds pre-treated with MT reduced ABA content and increased $GA_3$ content. Therefore, cotton seeds soaked with MT showed a significant improvement in the ability of seeds to germinate under PEG-6000 simulated drought stress. Our study highlights the beneficial effects of pre-treatment with MT on the germination of cotton seeds, as shown by multiple perspectives including morphological and physiological traits.

## ACKNOWLEDGEMENTS

The authors are grateful to the anonymous reviewers for their valuable comments and suggestions.

### Funding

This work was supported by the National Natural Science Foundation of China (No. 31871569), the Modern Technology System of Agricultural Industry in Hebei (HBCT2018040201, CL), and the Fund of Research Group Construction for Crop Science in Hebei Agricultural University (TD2016C318). The funders had no role in study design, data collection and analysis, decision to publish, or preparation of the manuscript.

## Grant Disclosures

The following grant information was disclosed by the authors:
National Natural Science Foundation of China: 31871569.
The Modern Technology System of Agricultural Industry in Hebei: HBCT2018040201, CL.
Research Group Construction for Crop Science in Hebei Agricultural University: TD2016C318.

## Competing Interests

The authors declare there are no competing interests.

## Author Contributions

- Yandan Bai conceived and designed the experiments, performed the experiments, analyzed the data, prepared figures and/or tables, authored or reviewed drafts of the paper, and approved the final draft.
- Shuang Xiao conceived and designed the experiments, prepared figures and/or tables, authored or reviewed drafts of the paper, and approved the final draft.
- Zichen Zhang performed the experiments, prepared figures and/or tables, authored or reviewed drafts of the paper, and approved the final draft.
- Yongjiang Zhang and Hongchun Sun analyzed the data, authored or reviewed drafts of the paper, and approved the final draft.
- Ke Zhang and Xiaodan Wang performed the experiments, authored or reviewed drafts of the paper, and approved the final draft.
- Zhiying Bai performed the experiments, prepared figures and/or tables, authored or reviewed drafts of the paper, and approved the final draft.
- Cundong Li conceived and designed the experiments, authored or reviewed drafts of the paper, and approved the final draft.
- Liantao Liu conceived and designed the experiments, analyzed the data, authored or reviewed drafts of the paper, and approved the final draft.

## Data Availability

The raw measurements are available as a Supplemental File.

## Supplemental Information

Supplemental information for this article can be found online at http://dx.doi.org/10.7717/peerj.9450#supplemental-information.

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
