# Peer review of "Melatonin improves the germination rate of cotton seeds under drought stress by opening pores in the seed coat"

_PeerJ, doi:10.7717/peerj.9450_

## Round 0.1 · original submission · Major Revisions

The reviewers made several suggestions on how to improve the manuscript. Importantly, several related articles are mentioned that you should consider, as well as inconsistencies in the reporting of data that you should correct.

Reviewer 1 ·

Basic reporting

It is okay.

Experimental design

It is okay.

Validity of the findings

It contains new findings on roles of melaotnin in plants.

Additional comments

This paper described the functional role of melatonin on cotton seed germination in response to drought stress. The data were similar to previous report of Simlat et al (2018), but contained advanced data. I just pointed out a couple of minor points prior to publication.

1. Title was somewhat misleading. Melatonin does not improve cotton germination, but improves germination under drought stress. Thus, add under drought stress in title.

2. line 146: delete an Inhibition and produce.

Reviewer 2 ·

Basic reporting

This is an interesting study on the use of melatonin for enhancing seed germination under drought stress in cotton. However, the introduction, methods and discussion need more detail. More recent literature should be cited. The English language should be improved. Some examples where the texts could be improved are listed below

Line 38-41: I would suggest to extend this section by detailing the global impacts of drought on plant physiology and crop production. More recent references (see below) should be cited.

Tomato WRKY81 acts as a negative regulator for drought tolerance by modulating guard cell H2O2–mediated stomatal closure. 2020. Environmental and Experimental Botany 171. doi:10.1016/j.envexpbot.2019.103960

Line 58: Antagonistic interaction between ABA and GA in regulating seed germination under stress conditions should be discussed in view of recent literature

Abscisic Acid and Gibberellins Act Antagonistically to Mediate Epigallocatechin-3-Gallate-Retarded Seed Germination and Early Seedling Growth in Tomato. 2020. Journal of Plant Growth Regulation. doi:10.1007/s00344-020-10089-1

Line 68-69: please do not generalize neurohormones and plant melatonin in terms of stress tolerance. This is misleading. I would suggest the authors to provide a general but specific introduction to melatonin in plants and animals. Also, describe in detail that plant melatonin or phytomelatonin plays crucial roles in both biotic and abiotic stress tolerance in plants in view of following literature

Melatonin alleviates iron stress by improving iron homeostasis, antioxidant defense and secondary metabolism in cucumber. 2020. Scientia Horticulturae 265. doi:10.1016/j.scienta.2020.109205

Alkanes (C29 and C31)-Mediated Intracuticular Wax Accumulation Contributes to Melatonin- and ABA-Induced Drought Tolerance in Watermelon. 2020. Journal of Plant Growth Regulation. doi:10.1007/s00344-020-10099-z

Role of melatonin in arbuscular mycorrhizal fungi-induced resistance to Fusarium wilt in cucumber. 2020. Phytopathology. doi:10.1094/phyto-11-19-0435-r


Line 135-141: Please describe the standards that you used to consider a seed germinated or not. I would suggest to provide more detail on the assessment of seed germination


Line 187: under the subheading “Determination of MT concentration”, it seems that authors would like to describe melatonin content in plants, but I do not find any description regarding endogenous melatonin content. This will mislead the readers. Please replace it with ‘Selection of MT concentrations for seed treatment’
Line 366-395: Discussion relating to ROS, oxidative stress and antioxidant potential as well as the role of plant growth regulators in improving stress tolerance could be further improved in view of the following literature. Authors are suggested to discuss and cite the following literature.

Abscisic Acid and Gibberellins Act Antagonistically to Mediate Epigallocatechin-3-Gallate-Retarded Seed Germination and Early Seedling Growth in Tomato. 2020. Journal of Plant Growth Regulation. doi:10.1007/s00344-020-10089-1

Dopamine alleviates bisphenol A-induced phytotoxicity by enhancing antioxidant and detoxification potential in cucumber. Environmental Pollution 259. doi:10.1016/j.envpol.2020.113957

Tomato WRKY81 acts as a negative regulator for drought tolerance by modulating guard cell H2O2–mediated stomatal closure. Environmental and Experimental Botany 171. doi:10.1016/j.envexpbot.2019.103960

Hasan MK, Ahammed GJ, Sun S, Li M, Yin H, Zhou J (2019) Melatonin Inhibits Cadmium Translocation and Enhances Plant Tolerance by Regulating Sulfur Uptake and Assimilation in Solanum lycopersicum L. Journal of Agricultural and Food Chemistry 67 (38):10563-10576. doi:10.1021/acs.jafc.9b02404

The English is understandable, but the quality of writing and technical construction of the narrative needs attention. The quality is almost good enough, but there are some atypical word choices, misplaced modifiers, vague language choices, and phrases. e.g. Line 426-: ‘pre-treating cotton seeds with MT’! There are many such examples throughout the manuscript.

Experimental design

Line 108-109; 168-171: Experiments were said to be conducted in 3-5 replicates for each experiment, but no information is presented for each of the repeats or how the experimental repeats where handled in the context of presented data.

Line 116: Please explain the basis for choosing 24 h duration as time-span of seed treatment.

Line 119-, 127-, 142-, 149-, 155-: Many important details need to be specified in “Materials and Methods”, such as (1) how the germination experiments were done (conditions, 7 days in dark?);(2) how the various samples were collected (including number of samples, sample units, replicates, tissue type (specify leaves/shoots/roots) detailed step-by-step protocol and more) for multiple tests, and ;(3) how exactly statistical analyses were performed.

Validity of the findings

Melatonin is a new and multifunctional molecule that confers tolerance to a number of biotic and abiotic stresses in plants. In the present manuscript, the authors studied the effects of seed treatment with melatonin on seed germination under drought stress in cotton.

Additional comments

This is an interesting study on the use of melatonin for enhancing seed germination under drought stress in cotton. However, the introduction, methods and discussion need more detail. More recent literature should be cited. The English language should be improved. Some examples where the texts could be improved are listed below

Line 38-41: I would suggest to extend this section by detailing the global impacts of drought on plant physiology and crop production. More recent references (see below) should be cited.

Tomato WRKY81 acts as a negative regulator for drought tolerance by modulating guard cell H2O2–mediated stomatal closure. 2020. Environmental and Experimental Botany 171. doi:10.1016/j.envexpbot.2019.103960

Line 58: Antagonistic interaction between ABA and GA in regulating seed germination under stress conditions should be discussed in view of recent literature

Abscisic Acid and Gibberellins Act Antagonistically to Mediate Epigallocatechin-3-Gallate-Retarded Seed Germination and Early Seedling Growth in Tomato. 2020. Journal of Plant Growth Regulation. doi:10.1007/s00344-020-10089-1

Line 68-69: please do not generalize neurohormones and plant melatonin in terms of stress tolerance. This is misleading. I would suggest the authors to provide a general but specific introduction to melatonin in plants and animals. Also, describe in detail that plant melatonin or phytomelatonin plays crucial roles in both biotic and abiotic stress tolerance in plants in view of following literature

Melatonin alleviates iron stress by improving iron homeostasis, antioxidant defense and secondary metabolism in cucumber. 2020. Scientia Horticulturae 265. doi:10.1016/j.scienta.2020.109205

Alkanes (C29 and C31)-Mediated Intracuticular Wax Accumulation Contributes to Melatonin- and ABA-Induced Drought Tolerance in Watermelon. 2020. Journal of Plant Growth Regulation. doi:10.1007/s00344-020-10099-z

Role of melatonin in arbuscular mycorrhizal fungi-induced resistance to Fusarium wilt in cucumber. 2020. Phytopathology. doi:10.1094/phyto-11-19-0435-r

Line 108-109; 168-171: Experiments were said to be conducted in 3-5 replicates for each experiment, but no information is presented for each of the repeats or how the experimental repeats where handled in the context of presented data.

Line 116: Please explain the basis for choosing 24 h duration as time-span of seed treatment.

Line 119-, 127-, 142-, 149-, 155-: Many important details need to be specified in “Materials and Methods”, such as (1) how the germination experiments were done (conditions, 7 days in dark?);(2) how the various samples were collected (including number of samples, sample units, replicates, tissue type (specify leaves/shoots/roots) detailed step-by-step protocol and more) for multiple tests, and ;(3) how exactly statistical analyses were performed.


Line 135-141: Please describe the standards that you used to consider a seed germinated or not. I would suggest to provide more detail on the assessment of seed germination


Line 187: under the subheading “Determination of MT concentration”, it seems that authors would like to describe melatonin content in plants, but I do not find any description regarding endogenous melatonin content. This will mislead the readers. Please replace it with ‘Selection of MT concentrations for seed treatment’
Line 366-395: Discussion relating to ROS, oxidative stress and antioxidant potential as well as the role of plant growth regulators in improving stress tolerance could be further improved in view of the following literature. Authors are suggested to discuss and cite the following literature.

Abscisic Acid and Gibberellins Act Antagonistically to Mediate Epigallocatechin-3-Gallate-Retarded Seed Germination and Early Seedling Growth in Tomato. 2020. Journal of Plant Growth Regulation. doi:10.1007/s00344-020-10089-1

Dopamine alleviates bisphenol A-induced phytotoxicity by enhancing antioxidant and detoxification potential in cucumber. Environmental Pollution 259. doi:10.1016/j.envpol.2020.113957

Tomato WRKY81 acts as a negative regulator for drought tolerance by modulating guard cell H2O2–mediated stomatal closure. Environmental and Experimental Botany 171. doi:10.1016/j.envexpbot.2019.103960

Hasan MK, Ahammed GJ, Sun S, Li M, Yin H, Zhou J (2019) Melatonin Inhibits Cadmium Translocation and Enhances Plant Tolerance by Regulating Sulfur Uptake and Assimilation in Solanum lycopersicum L. Journal of Agricultural and Food Chemistry 67 (38):10563-10576. doi:10.1021/acs.jafc.9b02404

The English is understandable, but the quality of writing and technical construction of the narrative needs attention. The quality is almost good enough, but there are some atypical word choices, misplaced modifiers, vague language choices, and phrases. e.g. Line 426-: ‘pre-treating cotton seeds with MT’! There are many such examples throughout the manuscript.

You need a little work on improving the coherency of the narrative as well as the English language so that it will be suitable to publish.

·

Basic reporting

No comment

Experimental design

No comment

Validity of the findings

No comment

Additional comments

Abstract
• Information regarding PEG should be given.
• Abbreviations for hydrogen peroxide, superoxide, superoxide dismutase, and peroxidase should be given.
• Which osmo-protectants were affected positively? It is unclear.
• The authors mentioned “Starch metabolism” in abstract, but there is no expression about amylase activity, GP, GR, etc.
• The abstract section should be re-written based on all parameters studies.

Introduction
• There is no information regarding amylase.
• The information regarding the other parameters studied should be extended.
• Line 43 .. Correct ref

Mat met
• Line 149: Soluble sugar, protein, amylase methods are not adequate. A detailed explanation should be given.
• The methods of ABA and GA contents must be given in detailed

Results
• Line 180 (PEG concentration) results must be checked. The values (%) are faulty.
• Line 219: The value of SOD is faulty.
• Similarly,
Line 225 (POD activity),
Line 236 (O2- content), Line 243
Line 251 (MDA) content
Line 261 (amylase content) values have been miscalculated.

• Line 269-270: The authors expressed that all treatments increased proline content; however, as seen from the figure, MT treatment did not statistically increased proline content on 2, 4, and 6th days.
• The names of treatments groups should be given as only abbreviation. The full names must be removed in this section.
• Fig 3 A: The unit of SOD activity must be given U.mg protein. U/g FW is not correct to discuss the results.
• Fig 4A : Change the unit of H2O2 content. gprotein expression is not correct.
Discussion
• The names of treatments groups should be given as only abbreviation. The full names must be removed in this section.
• The authors published a different study regarding effect of melatonin on germination of cotton seeds. In this study, they investigated amylase activity, proline, protein, sugar, H2O2, O2- contents and SEM different from prior study.
• The discussion section is not satisfactory. The results are interconnected and not fluent.
• The results are not well discussed, but simply passed over.
• Very little information is given about the pore opening.

---

## Round 0.2 · accepted · Accept

Thank you for your point-by-point response to the three reviewers' comments. Please consider proofreading the paper again to ensure that your work receives the attention it deserves; Besides, please check the use of 'Gibbereliilns' and 'Gibberellic acid'.

Reviewer 2 ·

Basic reporting

The revisions look fine. Professional article structure, figures, tables.

Experimental design

The revisions look fine. Research question well defined, relevant & meaningful.

Validity of the findings

All underlying data have been provided

Additional comments

I found that the authors adequately addressed my queries, although there are still some minor language issues in the revised texts. Perhaps the authors are still confused between Gibberellins and Gibberellic acid. Anyway, these minor issues could be solved during proof-correction, therefore, I would like to endorse the revised manuscript for publication.